# Post-abortion care with misoprostol – equally effective, safe and accepted when administered by midwives compared to physicians: a randomised controlled equivalence trial in a low-resource setting in Kenya

Marlene Makenzius,[1,2] Monica Oguttu,[3] Marie Klingberg-Allvin,[4,5] Kristina Gemzell-Danielsson,[6,7] Theresa M A Odero,[8] Elisabeth Faxelid[1]

The 31th International Confederation of Midwives (ICM), Toronto, June 2017

For numbered affiliations see end of article.

**Correspondence to**
Dr Marlene Makenzius;
marlene.makenzius@
folkhalsomyndigheten.se

## ABSTRACT

**Objective** To assess the effectiveness of midwives administering misoprostol to women with incomplete abortion seeking post-abortion care (PAC), compared with physicians.

**Design** A multicentre randomised controlled equivalence trial. The study was not masked.

**Settings** Gynaecological departments in two hospitals in a low-resource setting, Kenya.

**Population** Women (n=1094) with incomplete abortion in the first trimester, seeking PAC between 1 June 2013 to 31 May 2016. Participants were randomly assigned to receive treatment from midwives or physicians. 409 and 401 women in the midwife and physician groups, respectively, were included in the per-protocol analysis.

**Interventions** 600 µg misoprostol orally, and contraceptive counselling by a physician or midwife.

**Main outcome measures** Complete abortion not needing surgical intervention within 7–10 days. The main outcome was analysed on the per-protocol population with a generalised estimating equation model. The predefined equivalence range was −4% to 4%. Secondary outcomes were analysed descriptively.

**Results** The proportion of complete abortion was 94.8% (768/810): 390 (95.4%) in the midwife group and 378 (94.3%) in the physician group. The proportion of incomplete abortion was 5.2% (42/810), similarly distributed between midwives and physicians. The model-based risk difference for midwives versus physicians was 1.0% (−4.1 to 2.2). Most women felt safe (97%; 779/799), and 93% (748/801) perceived the treatment as expected/easier than expected. After contraceptive counselling the uptake of a contraceptive method after 7–10 days occurred in 76% (613/810). No serious adverse events were recorded.

**Conclusions** Treatment of incomplete abortion with misoprostol provided by midwives is equally effective, safe and accepted by women as when administered by physicians in a low-resource setting. Systematically provided contraceptive counselling in PAC is effective to mitigate unmet need for contraception.

### Strengths and limitations of this study

► Despite a restrictive abortion law the follow-up rate was considered high.
► A large sample size
► A rigorous project design and trial implementation was adopted in collaboration with Ministry of Health, different cadres of healthcare providers, leaders, NGOs and universities.
► District hospitals are usually more advanced regarding staffing and resources and might therefore not be representative of the small health dispensaries located in the most remote areas of the region and other parts of the country.
► Both midwives and physicians were working in the same health facility and therefore there could have been interaction between the two sets of providers in post-abortion care.
► The contraceptive up-take was high at follow-up (7–10 days), but lacks long-term follow-up.

**Trial registration number** NCT01865136; Results.

## INTRODUCTION

Although safe, simple and effective evidence-based interventions exist, about 22 million unsafe abortions take place each year worldwide.[1] An estimated 47 000 women die annually from complications resulting from resorting to unsafe practices for termination of pregnancy,[1] one of the leading causes of death among young women in sub-Saharan Africa.[1–6] Unsafe abortion is defined by the World Health Organization (WHO) as a procedure performed under conditions that fail to meet the minimal medical standards and/or by untrained persons.[3]

Access to post-abortion care (PAC) for treatment of the complications of unsafe abortion and spontaneous abortion (ie, all fetal or placental tissue have not been expelled and severe bleeding or infection occur, and prompt medical attention is required), has increased over decades.[1] However women in many countries still do not have access to this life-saving care or are mistreated when they seek it.[6] The costs for PAC constitute a significant financial burden on public healthcare systems.[1 7–9]

An estimated 4 65 000 induced abortions occurred in Kenya in 2012, 48 induced abortions in 1000 women of reproductive age (15–49 years) and in some regions 35% of maternal deaths are attributed to unsafe abortions.[2] Because of the restrictive abortion law in Kenya,[10] limited access to quality healthcare and stigma, most of the Kenyan abortions were done outside authorised facilities and thus considered unsafe.[2] An estimated 1 20 000 women were treated for complications resulting from unsafe abortion in 2012.[2] Moreover, the unmet need for family planning remains high in Kenya,[2 11 12] since 71% of women seeking PAC did not use a contraceptive method prior to becoming pregnant.[2 11] Contraception has clear health benefits as prevention of unintended pregnancies results in a subsequent decrease in maternal and infant mortality and morbidity.[11 13]

Even though there are no restrictions to treat abortion complications, most providers in Kenya have not been trained in PAC, in accordance with the WHO's technical and policy guidance.[3]

PAC is an integral component of comprehensive abortion care (CAC), and is often used in settings where abortion is legally restricted.[13 14] Essential elements are community and service provider partnership, counselling, treatment of incomplete and unsafe abortion, contraceptive services, and linkages to reproductive and other health services.[13] The prostaglandin $E_1$ analogue misoprostol is effective treatment for incomplete abortion in the first trimester, in PAC,[3 13–16] and no significant differences have been reported in the effectiveness between misoprostol and manual vacuum aspiration (MVA) for the treatment of incomplete abortion.[14 15] Misoprostol simplifies the procedure, compared with MVA, which is more intrusive.[13–16] Even though the use of MVA and misoprostol is increasing in Kenya, other less safe procedures remain common, reinforced by the mismatch between where maternal mortality occurs and where health workers are located.[17] In addition, regulations restrict misoprostol prescription and supervision to physicians, while nurses and midwives tend to have a subordinate role compared with physicians.[1 17]

Moving beyond specialist doctors to involve a wider range of health workers is an increasingly important public health strategy to reduce maternal mortality in low-resource settings.[18–22] Trained nurses and midwives have in some RCTs been shown to be able to provide safe medical abortion[23 24] and to treat incomplete abortion as effectively as physicians.[25] It is estimated that the global deficit of skilled healthcare professionals will reach 12.9 million by 2035.[26] Such shortages are especially critical in regions of the world that also have a high burden of unsafe abortion and related mortality. Thus expanding health worker roles opens a window for improving access to safe PAC.[20 26] The range of safe and effective options, including non-surgical methods, can facilitate evidence-based decision-making and adaptation to the local context, resources and public health needs.[1 26–29] The aim of this trial was to assess the effectiveness of midwives diagnosing incomplete abortion and administering misoprostol to women needing PAC, compared with physicians.

## METHODS

The study was a multicentre randomised parallel-group trial, reported in accordance with the CONsolidated Standards of Reporting Trials (CONSORT) guidelines for non-inferiority and equivalence randomised trials.[30] The project was conducted at the Jaramogi Oginga Odinga Teaching and Referral Hospital (JOOTRH) and Kisumu County Hospital (KCH) in Kisumu County, western Kenya. Regional variations in abortion incidence exist in Kenya, and the rate in this region (Nyanza/Western) was among the highest – 63 induced abortions per 1000 women of reproductive age (15–49 years), compared with 48 for the country.[2 17] Consequently, severe complications from unsafe abortion were common, and associated with delays in seeking healthcare being managed in a lower- level facility, and being referred to an upper-level facility.[17] The superior regime to treat first trimester incomplete abortion was MVA (62%), similar to the country (65%), and medical abortion catered for only 11%.[17] In the region about 37% of women (15–49 years) had never used a contraceptive method compared with 46% in the country.[17]

Kisumu is the third largest city in Kenya and has a population of about 5 00 000 people. The project was implemented in the gynaecological ward within the Department of Obstetrics and Gynaecology at the two facilities (1 June 2013 to 31 May 2016). Altogether the two facilities admit between 20–32 women with incomplete abortions per month.

The study was designed and coordinated by researchers at the Department of Public Health Sciences, Global Health (IHCAR), and Women's and Children's Health, Karolinska Institutet, Stockholm, Sweden; Kisumu Medical Education Trust (KMET); University of Nairobi; Moi University, Eldoret, Kenya; and at JOOTRH and KCH in Kisumu, Kenya.

Research questions and outcome measures were constructed by the researcher team and healthcare providers at the study facilities who were experienced in PAC, in order to strengthen the validity. The preliminary results was continuously reported during the study period to the two facilities by the study coordinator to encourage the provider's work, who also used the information to encourage the participating women. The outcome and acceptability of the intervention was evaluated by the

research assistants at follow-up, and the women were advised to seek care ahead of the appointment if they experienced unexpected symptoms/burden.

We obtained ethical approval from the JOOTRH Ethical Review Committee (diary number ERC42/13) and the Swedish regional Ethics Committee in Stockholm (2013/902-31/1). All the patients gave written informed consent. The participants could withdraw from participation at any time without consequences. Consent forms were translated into the local languages Kiswahili and Luo, and translated back into English to ensure accurate translation. The form was read to each participant and those unable to write marked the form with a thumb print, and a witness signed to indicate that the study had been explained to the participant and that she understood and accepted to participate.

## Inclusion and exclusion criteria

Inclusion criteria were women with signs of incomplete abortion, based on clinical assessments (ie, positive pregnancy test, bleeding, contractions and lower abdominal pain, open cervical os and partial expulsion) and physical examination (including vaginal examination), but without the use of ultrasound. Exclusion criteria were complete abortion (no bleeding or contractions), a uterine size estimated over 12 weeks of gestation, suspected ectopic pregnancy (severe abdominal pain), unstable haemodynamic status, signs of pelvic infection or sepsis, and known allergy to misoprostol.

## Randomisation and masking

Eligible women who consented to participate were randomly allocated to a midwife (intervention) or standard care with a physician (control) for diagnosis and treatment. The randomisation was done in blocks of eight and was stratified for study site. We used a computer random number generator to generate a list of codes from 1 to 890 and each code was linked to one of the two study groups. Sequentially numbered, opaque, sealed envelopes, each containing a random allocation, were prepared at the coordinating department of KMET, and later opened in consecutive order by the research assistants after obtaining written consent. Data management was organised locally at the coordinating department at KMET, Kisumu.

The study coordinator checked case report forms for accuracy, and corrected them if needed after discussion with research assistants, and did continuous process evaluation. The providers received support and guidance throughout the study period by the study coordinator. The study was not masked to the study participants or providers.

## Procedures

Eligible providers for participation were midwives (n=19) and physicians (n=18) (table 1), involved in PAC at the two facilities. The providers were trained by the research team in a 5-day training programme according to the Kenyan

**Table 1** Background characteristics of participating PAC providers

| Characteristics of providers | Midwife n=19 (51.4%) | Physician n=18 (48.6%) | Total n=37 (100%) |
|---|---|---|---|
| Age | | | |
| Mean (SD) | 47.3 (6.5) | 35.9 (10.0) | 41.7 (10.1) |
| Median | 50 (6.0) | 30 (14.5) | 44 (20) |
| Range | 28–55 | 26–60 | 26–60 |
| Sex | | | |
| Female | 18 (94.7%) | 6 (33.3%) | 24 (64.9%) |
| Male | 1 (5.3%) | 12 (66.7%) | 13 (35.1%) |
| Years of professional practice | | | |
| Mean (SD) | 22.4 (7.9) | 8.8 (8.1) | 15.8 (10.5) |
| Median (IQR) | 26 (8.0) | 4 (15) | 17 (22) |
| Range | 1–30 | 1–28 | 1–30 |
| Clinical experience in PAC (years)* | | | |
| Mean (SD) | 2.7 (1.6) | 8.4 (6.9) | 5.5 (5.5) |
| Median (IQR) | 3 (3.0) | 4 (4.25) | 3 (2.5) |
| Range | 0–5 | 0–25 | 0–25 |

Data are n (%), mean (SD) or median (IQR).
*Before study start.

and the standardised PAC training module.[3 31 32] The training was facility-based and both theoretical and practical, and was repeated twice during the project period in order to cover all/new staff. This training included knowledge on diagnosing incomplete abortions, treatment with misoprostol and MVA, and post-abortion contraceptive counselling as well as interview technique. The healthcare facilities were equipped by the project to provide basic and emergency obstetric services and were estimated to have sufficient staff and resources required for the trial. The procedure and the instruments were pilot-tested at the two hospitals before the study started. The first 2 months was set to a run-in phase, in which the facilities were given time to set the routine among staff for screening, enrollment and follow-up, as well as to free space to store a large number of protocol-records safely, and to provide counselling in a secluded room. Minor revisions were made to the instruments and the recruitment procedure.

Out of the 19 midwives, four per facility were trained as research assistants in the study. They were responsible for overseeing and monitoring supply in stock (misoprostol and contraceptives such as condoms, pills, Depo-Provera (medroxyprogesterone acetate injectable), hormonal implants and intra uterine devices (IUD)). In addition, the research assistants were responsible for the follow-up visit including assessment of treatment outcome and contraceptive uptake, and performing MVA when necessary.

Women admitted with signs of incomplete abortion were first screened by a research assistant, based on

self-reported last menstrual period, pregnancy test and symptoms. Eligible women who consented to participate were randomly allocated to the midwife or physician group. The clinical assessment included (table 2): the last menstrual period, obstetric and gynaecological history, pelvic examination including any signs of genital infections, cervical status, bleeding and size of the uterus. Ultrasound was not used.

Each participant was given one single dose of 600 µg misoprostol orally. Additionally, they were offered analgesics (ibuprofen or paracetamol) and oral antibiotics given according to local guidelines for PAC. For monitoring purposes, participants were advised to stay at the clinic for 4 to 12 hours after swallowing the misoprostol tablets. Before discharge, all women were offered contraceptive counselling and provided with a contraceptive method if they wanted, and given a follow-up appointment after 7–10 days.

Participants received detailed information about bleeding, pain expected, abnormal symptoms after treatment (severe pain, fever and foul-smelling vaginal discharge), and were informed about the importance of seeking care or consulting on the toll-free line open 24/7 if such symptoms occurred. Side-effects were recorded (online supplementary appendix 1). The women were offered reimbursement for travel costs as an encouragement to come for follow-up. All women received an SMS reminder 2 days before the scheduled follow-up appointment. Women who did not show up were contacted by the research assistant using a cell phone provided by the project, and the woman was given a new appointment and encouraged to attend.

## Outcomes

The pre-specified primary outcome was complete abortion, requiring no further medical or surgical intervention (table 3). A symptom diary card was used to assess secondary outcomes following the treatment (table 4).
i. daily bleeding in relation to normal menstrual bleeding,
ii. pain following the treatment,
iii. unscheduled visits at a healthcare facility and the reason,
iv. women's acceptability of the treatment was recorded by three questions 'How did you perceive the treatment procedure'; 'Did you feel safe after the treatment'; and 'Will you recommend the treatment to a friend or a relative'

Contraceptive uptake (part of comprehensive PAC) was an additional outcome recorded by three questions 'Did you receive contraceptive counselling'; 'Did you accept a contraceptive method'; and 'What method did you chose' (table 5).

## Statistical analyses

The sample size was calculated with the objective of showing two-sided equivalence assuming that the rate of incomplete abortions could be 4% and would apply to both types of providers.[26 29] A pre-defined acceptable difference in completion rate between the two providers ranged from –4% to 4%. The margin was based on previous studies and clinically and statistically important differences as well as ethical criteria, cost and feasibility.[24 26 30] To establish equivalence with a power of 80% and two-sided 95% CI, the required sample size was 400 per group. Compensation for about 10% loss to follow-up gave a total sample size of 888. Background characteristics were analysed and presented with descriptive statistics, as per the intention to treat, that is, patients correctly allocated to study group and having received treatment. Main and secondary outcomes were analysed as per-protocol, since they were measured at follow-up.

Analysis of the primary outcome (complete abortion) was made with a generalised estimating equation model to estimate the risk difference between the two providers with two-sided 95% CI. If the CI of the risk difference between the two groups fell within the predetermined margin of equivalence (–4% to 4%), the two types of providers were considered equivalent. P values of 0.05 or less were considered statistically significant. Additionally, we adjusted for the following covariates: facility, age (<25 vs ≥25 years), marital status (single vs married or cohabiting), education (none, primary 1–4, or 5–8 vs secondary or tertiary), number of previous pregnancies (1 vs >1) and live births (0 vs ≥1). We estimated the adjusted risk difference as the predicted risk difference at the average of all included covariates. Safety parameters of the procedure were viewed descriptively without any formal statistical testing. The study protocol was not changed after the run-in period. All data were entered continuously throughout data collection and cleaned in IBM SPSS Statistics for Windows version 22.0, and we did all the statistical analysis within this software. The trial is registered at ClinicalTrials.gov, number NCT01865136.

## Role of the funding source

The authors designed this as an investigator-initiated trial, in collaboration with healthcare providers and management at the study setting. The funders did not play any role in the study design or in the collection, analysis and interpretation of data. All authors had full access to the data, and the first author regularly reported preliminary results to the whole research team. The corresponding author had final responsibility for the decision to submit the manuscript for publication.

## RESULTS

From 1 June 2013 to 31 May 2016, we assessed 1094 women with signs of incomplete abortion for eligibility for participation in this study (figure 1). Finally, a total of 810 women, 409 in the midwife group and 401 in the physician group, received misoprostol treatment and were followed-up. All those were included in the per-protocol analysis of the main outcome.

**Table 2** Sociodemographic characteristics and reproductive history of participants* by provider

| Characteristics of participants | Midwife | Physician | Total |
|---|---|---|---|
| **Age (years)** | | | |
| n | 422 | 416 | 838 |
| Mean (SD) | 25.2 (5.9) | 24.8 (5.4) | 25- (5.6) |
| Range | 14–45 | 12–41 | 12–45 |
| 14–19 | 60 (14%) | 56 (14%) | 116 (14%) |
| 20–24 | 171 (40%) | 167 (40%) | 338 (40%) |
| 25–29 | 101 (24%) | 113 (27%) | 214 (26%) |
| 30–34 | 53 (13%) | 51 (12%) | 104 (12%) |
| 35–39 | 28 (7%) | 26 (6%) | 54 (7%) |
| 40–47 | 9 (2%) | 3 (1%) | 12 (1%) |
| **Marital status** | | | |
| n | 433 | 426 | 859 |
| Married or cohabiting | 289 (67%) | 292 (68%) | 581 (68%) |
| Single | 128 (30%) | 124 (29%) | 252 (29%) |
| Divorced/separated | 9 (2%) | 7 (2%) | 16 (2%) |
| Widowed | 7 (2%) | 3 (1%) | 10 (1%) |
| **Religion** | | | |
| n | 433 | 426 | 859 |
| Christian | 427 (99%) | 417 (98%) | 844 (98%) |
| Muslim | 6 (1%) | 9 (2%) | 15 (2%) |
| **Education** | | | |
| n | 433 | 425 | 858 |
| None | 6 (1%) | 2 (1%) | 8 (1%) |
| Primary 1–4 | 11 (2%) | 9 (2%) | 20 (2%) |
| Primary 1–5 | 108 (25%) | 121 (28%) | 229 (27%) |
| Secondary | 206 (48%) | 186 (44%) | 392 (46%) |
| Tertiary | 102 (24%) | 107 (25%) | 209 (24%) |
| **Occupation** | | | |
| n | 431 | 423 | 854 |
| Unemployed | 210 (49%) | 197 (47%) | 407 (48%) |
| Formal employment | 73 (17%) | 75 (18%) | 148 (17%) |
| Self-employed | 148 (34%) | 151 (36%) | 299 (35%) |
| **Gestational age based on clinical examination (weeks)** | | | |
| n | 433 | 426 | 859 |
| Mean (SD) | 9.4 (2.2) | 9.7 (2.1) | 9.6 (2.2) |
| Range | 1–12 | 3–12 | 1–12 |
| **Number of previous pregnancies** | | | |
| n | 433 | 425 | 858 |
| Mean (SD) | 1.75 (1.8) | 1.8 (1.6) | 1.8 (1.7) |
| Range | 0–12 | 0–8 | 0–12 |
| 0 | 135 (31%) | 101 (24%) | 236 (27%) |
| 1 | 94 (22%) | 110 (26%) | 204 (24%) |
| 2–3 | 136 (31%) | 157 (37%) | 293 (34%) |
| 4–5 | 58 (13%) | 50 (11%) | 108 (13%) |

| Table 2 | Continued | | |
|---|---|---|---|
| **Characteristics of participants** | **Midwife** | **Physician** | **Total** |
| 6–8 | 7 (2%) | 7 (2%) | 14 (2%) |
| 9–12 | 3 (1%) | 0 | 3 |
| Parity (live birth) | | | |
| n | 431 | 425 | 856 |
| Mean (SD) | 1.1 (1.4) | 1.1 (1.3) | 1.1 (1.3) |
| Range | 0–10 | 0–7 | 0–10 |
| 0 | 205 (48%) | 178 (42%) | 383 (45%) |
| 1 | 105 (24%) | 111 (26%) | 216 (25%) |
| 2–3 | 94 (22%) | 115 (27%) | 209 (24%) |
| 4–5 | 23 (5%) | 18 (4%) | 41 (5%) |
| 6–8 | 3 (1%) | 3 (1%) | 6 (1%) |
| 9–11 | 1 | 0 | 1 |

Data are n (%) unless otherwise stated.
Internal drop-out range between 0–11 (0%–2.6%), among the midwife and physician group, respectively.
*Includes women lost to follow-up.

Among the providers, the midwives had longer work experience than the physicians. On the other hand, the physicians were more experienced in PAC than the midwives in PAC (table 1).

The baseline characteristics such as sociodemographic background and reproductive history were similarly distributed between the two groups (table 1). The mean duration of gestational age based on clinical examination was 9.6 weeks (range 1–12 weeks; table 2).

Sub-analyses showed slightly higher proportions of young (<25 years), single, unemployed and nulliparous women among those who dropped out of the study compared with those who came for the follow-up visits. There were no differences in gestational age or education levels between the two groups. None of the results of the sub-analyses were statistically significant (p>0.05).

### Main outcome
The overall proportion of complete abortion was 94.8% (768/810). As shown in table 3, 390/409 (95.4%) women in the midwife group and 378/401 (94.3%) women in the physician group had complete abortions. The model-based risk difference for midwife versus physician group was 1.0%

(–4.1 to 2.0). There were 42/810 incomplete abortions (5.2%), 19/409 (4.6%) in the midwife group and 23/401 (5.7%) in the physician group. Among the 42 women with incomplete abortions, 25 were treated with MVA and 17 were treated with a repeat dosage of misoprostol after they completed the follow-up assessment. Among the 25 women who obtained MVA, two were treated with a repeat dosage of misoprostol prior to MVA. No serious adverse events were recorded.

### Secondary outcomes
Table 4 shows that most women (690; 86%) reported bleeding less than, or the same as, normal menstrual bleeding after treatment, and the mean number of bleeding days was 4.1 (SD 2.0; range, 1–14). A majority (564/806; 70%) of the women reported no or mild pain after treatment, and 6% (46/806) women reported unscheduled visits. Vaginal bleeding and abdominal pain were reported as reasons for the unscheduled visits. Most women perceived the treatment to be as expected or easier than expected (748/801; 93%), and 97% felt safe during the procedure (779/799). Additionally, 95% (761/799) women stated that they would recommend the treatment to a friend. There were no significant

| Table 3 | Outcome of treatment of incomplete abortion by provider | | | |
|---|---|---|---|---|
| | **Midwife** | **Physician** | **Risk difference (95% CI)** | **Adjusted difference (95% CI)*** |
| Randomised and received intervention | 446 | 444 | | |
| Per-protocol | 409 | 401 | | |
| Complete abortion | 390 (95.4%) | 378 (94.3%) | 1.0% (–4.1 to 2.0) | 1.0% (–4.1 to 2.2) |

*Adjusted for facility, age (<25 vs≥25 years), marital status (single, widowed, divorced, separated] vs married or cohabiting, education (none or primary 1–8 vs secondary or tertiary), number of previous pregnancies (0 vs >1), and parity (para 0 vs ≥1).

**Table 4** Secondary outcomes, symptoms following the treatment, by provider

| Secondary outcomes | Midwife | Physician | Total |
|---|---|---|---|
| **Bleeding since treatment** | | | |
| **N** | 408 | 401 | 806 |
| Less than normal menstrual bleeding | 274 (67%) | 268 (67%) | 542 (67%) |
| Same as normal menstrual bleeding | 78 (19%) | 70 (18%) | 148 (19%) |
| Heavier than normal menstrual bleeding | 56 (14%) | 60 (15%) | 116 (14%) |
| Days bleeding | | | |
| N | 396 | 389 | 785 |
| Mean | 4.2 (2.0) | 4.08 (1.7) | 4.1 (2.0) |
| Range | 1–14 | 1–10 | 1–14 |
| Pain following treatment | | | |
| N | 407 | 399 | 806 |
| None | 86 (21%) | 84 (20%) | 170 (21%) |
| Mild | 200 (49%) | 194 (49%) | 394 (49%) |
| Moderate | 81 (20%) | 76 (19%) | 157 (20%) |
| Severe | 40 (10%) | 45 (11%) | 85 (10%) |
| Unscheduled visit | | | |
| N | 403 | 397 | 800 |
| Yes | 20 (5%) | 26 (6%) | 46 (6%) |
| Vaginal bleeding or abdominal pain, or both | 11 (3%) | 18 (4%) | 29 (4%) |
| Chill/fever or dizziness, or both | 6 (1%) | 6 (1%) | 12 (1%) |
| Acceptance of the method | | | |
| How did you perceive the treatment | | | |
| N | 407 | 398 | 801 |
| As expected | 121 (30%) | 130 (32%) | 251 (31%) |
| Worse than expected | 29 (7%) | 27 (7%) | 56 (7%) |
| Easier than expected | 257 (63%) | 241 (61%) | 498 (62%) |
| Felt safe | | | |
| N | 400 | 399 | 799 |
| Yes | 383 (96%) | 378 (95%) | 779 (97%) |
| No | 17 (4%) | 21 (4%) | 38 (3%) |
| Will recommend the treatment to a friend | | | |
| N | 400 | 399 | 799 |
| Yes | 383 (96%) | 378 (95%) | 761 (95%) |
| No | 17 (4%) | 21 (5%) | 38 (5%) |

Data are n (%) unless otherwise stated.
Internal drop-out range between 0–11 (0%–2.6%), among the midwife and physician group, respectively.

(p<0.05) differences in the secondary outcomes between the groups. 407/810 women reported no side-effects following the treatment, while 402/810 women reported symptoms; abdominal pain, chills, nausea, diarrhoea, vomiting, and foul-smelling vaginal and/or cervical discharge, and were similar in both groups (online supplementary appendix 1).

## Contraceptive up-take

Contraceptive counselling was provided to 789 of the 810 (98%) women. Of these, 613 (76%) accepted a contraceptive method of their own choice (table 5). There were no significant differences in the contraceptive up-take between the study groups (p=0.342). The following choices of contraceptive methods were offered, and similarly distributed between the two types of providers: injectable (236/613; 39%), pills (166/613; 27%), condoms (151/613; 25%), hormonal implant (45/613; 8%), IUD (8/613; 1%): and permanent contraception (1/613; 0.16%).

**Table 5** Contraceptive uptake, by provider type

| Contraceptive uptake | Midwife | Physician | Total |
|---|---|---|---|
| Received contraceptive counselling | | | |
| N | 405 | 401 | 806 |
| Yes | 395 (98%) | 394 (98%) | 789 (98%) |
| No | 10 (2%) | 7 (2%) | 17 (2%) |
| Accepted a contraceptive method | | | |
| N | 407 | 401 | 803 |
| No | 104 (26%) | 91 (23%) | 195 (24%) |
| Yes | 303 (74%) | 310 (77%) | 613 (76%) |
| What method did you chose | | | |
| Injectable | 118 (39%) | 118 (38%) | 236 (39%) |
| Pills | 81 (27%) | 85 (28%) | 166 (27%) |
| Condoms | 75 (25%) | 76 (25%) | 151 (25%) |
| Hormonal implants | 24 (8%) | 21 (7%) | 45 (8%) |
| IUDs | 3 (1%) | 5 (2%) | 8 (1%) |
| Permanent contraceptives | – | 1 | 1 |

Data are n (%) unless otherwise stated.
Internal drop-out range between 0–11 (0%–2.6%), among the midwife and physician group, respectively.

## DISCUSSION

The high proportion of complications related to unsafe abortion continues to pose a major public health challenge in Kenya.[17] Our finding shows that treating women with incomplete abortion (≤12 weeks of gestation), with misoprostol was equally effective and safe when provided by midwives or physicians in a low-resource setting in Kenya. In addition, the treatment is highly accepted among the treated women, and three out of four women chose to start a contraceptive method, when contraceptive counselling is systematically provided in post-abortion care (PAC).

The overall complete abortion rate was 94.8%, which is comparable with the results from other studies that have used the same regimen (94.5%–96.7%).[16 21 24 25] The result emphasises that the safety of using misoprostol for treatment of incomplete abortion in low-resource settings, can be performed by trained midwives without the routine use of ultrasonography before or after abortion.

Of the women with incomplete abortion (42/810), 25 were treated with MVA and 17 with repeated dosage of misoprostol, similarly distributed between the two provider types. Repeat dosage of misoprostol is apparently an alternative for MVA when the first dosage of misoprostol has failed. This could be compared with results from a similar study from Uganda where all women with incomplete abortion were treated with MVA (36/955).[25]

A notable additional finding was the frequency of women with sepsis, which was a criterion for ineligibility in this study (47/1094). This proportion is higher than

has been seen in other African countries.[2 33] This might be an important reflection of the magnitude of unsafe abortion and the immense poor state of reproductive health access for women in Kenya. There is no doubt that there is room for improving timely and accessible CAC services, in particular PAC, in Kenya.

Although these results clearly show that misoprostol is a simple, efficient and desirable method, which is highly accepted by women, implementation of PAC in Kenya is hampered by the lack of national clinical guidelines.[2 33] Laws and policies governing access to safe abortion in Africa vary, ranging from very restrictive, such as in Kenya, where abortion is permitted only to save the life of the mother, to liberal, such as in South Africa, where abortion is permitted through the 12 weeks of pregnancy or later.[1] Stigmatising attitudes and beliefs surrounding abortion in the society, not least among healthcare providers,[20 29] contribute to a delay in women seeking timely care for abortion-related complications.[2]

Kenya has a policy framework to prevent maternal mortality and morbidity. However, this policy framework excludes unsafe abortion, which is addressed through the provision of PAC.[2 17] In 2012, the Ministry of Health (MH) developed clinical guidelines on prevention of morbidity and mortality from unsafe abortions in Kenya, but the document was arbitrarily withdrawn in 2013. A task force was commissioned by the Director of Medical Services to consult and revise the guidelines, and the work was finalised in September 2015, but since then, signing of the document has stalled. Simultaneously, women seeking treatment for complications arising from unsafe abortions is on the rise.[1 2]

New guidelines may not only contribute to increase the quality and effectiveness to treat complications of unsafe abortion, it may also encourage the prevention of unintended pregnancy. Unmet contraceptive need and contraceptive failure will consistently be associated with high rates of unplanned and unwanted pregnancies, forcing many women to resort to unsafe abortion with a high rate of complications including infertility, long-term morbidity and death.[33] The current result shows that, systematically provided, contraceptive counselling to PAC-seeking women is feasible and effective, as three out of four women accepted a contraceptive method. However, the use of modern LARC methods, such as hormonal implants and IUDs, was extremely low, in comparison to injectables. This indicates a pronounced need for concerted efforts to urgently reach women who have an unmet contraceptive need by giving them access to post-abortion contraceptive counselling and modern methods. Mitigating the unmet need for contraception could be the key to reducing the numbers of subsequent unintended pregnancies and unsafe abortions, as well as maternal morbidity and mortality.

The strengths of the study lie in its large sample size, high follow-up rate and the socio-demographic characteristics of the women (ie, heterogeneous, though overall low socioeconomics similar to those of the population

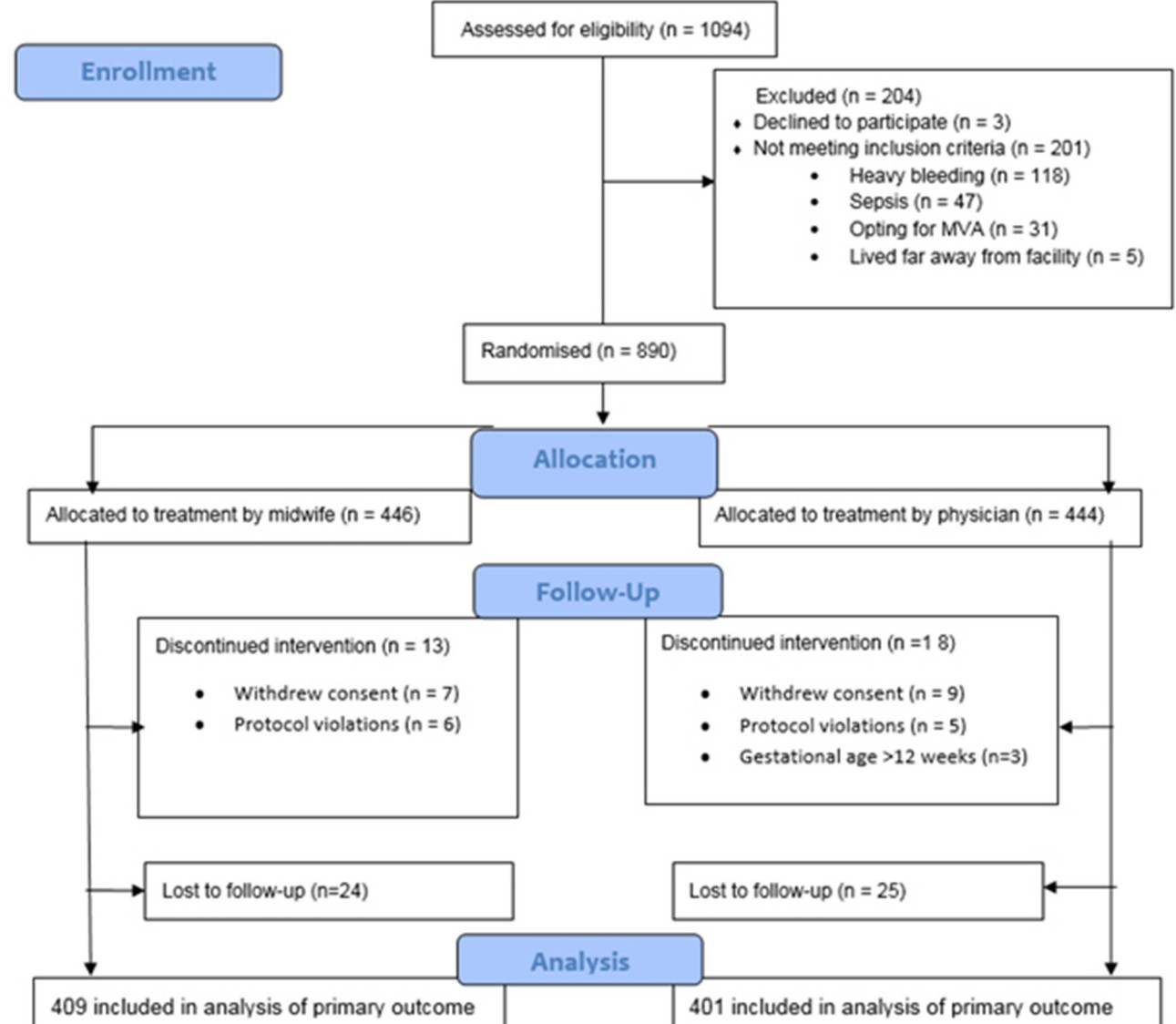

**Figure 1** Trial profile.

in the country.[17] There are however weaknesses in the study to be considered. First, the midwives had more years of professional experience than the physicians, but the physicians were more experienced in PAC. No more information was given in detail about their working experience, however, all the providers within the two facilities received the same training in the treatment of incomplete abortion. Second, although the socioeconomic and demographic characteristics of study participants were representative of women iof fertile ages in Kisumu County, they might not be representative of women living in the most remote rural areas of this county. Third, district hospitals are usually more advanced regarding staffing and resources and might therefore not be representative of the small health dispensaries located in remote areas of the region. Fourth, both types of providers worked in the same hospital and in the same environment. We cannot rule out interactions between the two provider groups, which might have led to a convergence of outcome. Fifth,

drop-outs to follow-up was a challenge, as well as the lack of long-term follow-up for the contraceptive up-take.

Though we encouraged women to come for follow-up by paying for the local transport, sent SMS reminders ahead of the scheduled appointment and called the patient if they did not show up, 49 of the 859 women did not come for the follow-up. Some of the women did not have a phone and were thus not reachable, and some of the women who were contacted by phone did not find it necessary to attend the follow-up, as they considered themselves to have recovered without any side effects. In addition, the adverse event profile matches the known adverse profile of the drug and there were no striking differences between the two categories of providers in this regard. Loss to follow-up could also be related to abortion stigma, which is present within this setting and other settings where abortion is restricted.[20] Women seeking PAC have previously reported shame and guilt, and they don't want to be associated with the abortion clinic and its activities

once the abortion issue is completed.[20][29] Therefore, in coherence with Osur et al. (2013), instead of considering drop-outs criteria as bias, this might rather be an important reflection of the stigma surrounding abortion.[20]

Future research needs to investigate if the results of this study are replicable in rural primary healthcare settings, such as health centres and dispensaries since it may differ from healthcare at district level. Moreover, future research needs to address long-term follow-up on contraceptive adherence after PAC. In spite of the convincing rationality in PAC of preventing future unplanned pregnancies by providing post-abortion contraceptive services, this component remains one of the weakest parts of post-abortion care.[1][2][13][33]

In conclusion, treatment of incomplete abortion with misoprostol when provided by midwives is equally effective, safe and accepted as when administered by physicians in a low-resource setting. Systematically provided contraceptive counselling composed with a wide range of contraceptive choices in PAC is effective to mitigate unmet need for contraception. In areas with shortages of physicians and scarce resources, midwives would greatly increase women's access to safe and highly accepted PAC. These findings are timely and are needed to accomplish standards and guidelines regarding CAC and PAC in Kenya and in other African countries lacking such strategies.

**Author affiliations**
[1]Department of Public Health Sciences Global Health (IHCAR), Karolinska Institutet, Stockholm, Sweden
[2]Department of Public Health and Caring Sciences, Uppsala University, Uppsala, Sweden
[3]Kisumu Medical and Education Trust (KMET), Reproductive Health, Kisumu, Kenya
[4]School of Education, Health and Social Studies, Dalarna University, Falun, Sweden
[5]Department of Women´s and Children´s Health, Karolinska Institutet, Stockholm, Sweden
[6]Division of Obstetrics and Gynaecology, Karolinska University Hospital, Stockholm, Sweden
[7]Departement of Women´s and Children´s Health, Karolinska Institutet, Stockholm, Sweden
[8]College of Health Sciences, School of Nursing Sciences, University of Nairobi, Nairobi, Kenya

**Acknowledgements** The authors acknowledge Caroline Nyandat and Sam Owoko for training and providing technical support, the data assistants Christine Apondi and Beatrice Otieno, the staff of the Ministry of Health in Kisumu, Dr Paul Mitei, and Professor Edwin Were, Moi University, who made this project possible and facilitated its implementation. Thanks to Ulla Romild at the National Agency of Public Health, Sweden, for statistical advice. Finally, we thank all the women who participated in our study, and all the providers involved in the study who remained committed and supportive, throughout.

**Contributors** The lead author for this manuscript was MM. Study design and planning: KG Danielsson, EF, Marie Klingberg Allvin and MO. Training and implementation of the study: Marie Klingberg Allvin, MO, KG Danielsson, EF, Theresa Odero and MM. Data collection: MM, MO and Theresa Odero. Data analysis: MM, cross-checked by a senior statistician, Ulla Romild, at the National Agency of Public Health in Sweden. Data interpretation: MM, EF, KG Danielsson, Marie Klingberg Allvin, MO and Theresa Odero. Literature search: MM. Writing: MM. Editing and proofreading: KG Danielsson, EF, Marie Klingberg Allvin, MO, Theresa Odero and Scribendi (language editing).

**Funding** The Swedish Research Council (2012-06114; 2011-03369), The Swedish Research Council on Health, Working Life and Welfare (2015-01398_3).

**Competing interests** None declared.

**Patient consent** Obtained.

**Ethics approval** JOOTRH Ethical Review Committee (diary number ERC42/13), and the Swedish regional Ethics Committee in Stockholm (2013/902-31/1).

**Provenance and peer review** Not commissioned; externally peer-reviewed.

**Data sharing statement** Additional data is available by emailing marlene.makenzius@ki.se.

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
