## [Reviewer comments · BMJ Open]

ARTICLE DETAILS

TITLE (PROVISIONAL)	Post-abortion care with misoprostol - equally effective, safe and accepted when administered by midwives compared to physicians: a randomised controlled equivalence trial in a low-resource setting in Kenya
AUTHORS	Makenzius, Marlene; Oguttu, Monica; klingberg-allvin, marie; Gemzell, Kristina; Odero, Theresa; Faxelid, Elisabeth

VERSION 1 – REVIEW

REVIEWER	Sanjoy Kumar Bhattacharyya R G Kar Medical College Kolkata India
REVIEW RETURNED	17-Feb-2017

GENERAL COMMENTS	Title: The title of the manuscript needs to be a bit informative. Misoprostol can be applied by different routes while managing incomplete abortion. Among them, the authors here had chosen the oral route and this need to be reflected in the study-title. Introduction: The introduction part is very descriptive, elaborate as well as over-informative, which are often a bit unrelated with the purpose of the study. Yes, incomplete abortion is a very commonly encountered problem and possibility of a medical management of this complication by ground level healthcare provider is of immense help. With this background, a short, brief and informative introduction is needed. Methods: I have definitely few major issues regarding the method-section of this study. 1. What are the diagnostic criteria of "incomplete abortion"? The authors have included "women with signs of incomplete abortion (i.e., bleeding, contractions and lower abdominal pain during pregnancy, open cervical os, and partial expulsion)". But these signs are never alone can diagnose even a pregnant state if you do not do a simple urine for pregnancy testing initially. Moreover, this is often difficult to differentiate between complete abortions with incomplete abortion even after clinical examination. But authors here do not take the help of ultrasonography before enrolling. These points need to clarify here in details. 2. What were the basic differences in enrollment, intervention and follow-up in between two groups, while managing the incomplete abortions?
--

	Are the diagnosis and subsequent enrollment of women was done by the physician or midwife in their respective group [So that there could be a more specific enrollment which could avoid the possibility of inclusion of lots of complete abortions]? [ ] No. Enrollment was done by two midwives initially irrespective of subsequent allocation. Thus, in this respect, these two groups are same. Are the basic intervention is technically difficult or need expertise or experience so that it might create a difference in between the groups? No. Only a single dose of oral Misoprostol was applied in both groups and this needs neither physician's knowledge nor midwife's experience. Were the follow up done by the groups differently? Not mentioned in details. This is a very important area of this study where you need to ensure whether the woman has achieved the complete abortion. How it was ensured? The authors have mentioned few signs of untoward events and absence of them is considered as the successful outcome. But this needs to be more detailed. Moreover, who has done those follow-up? Respective groups or any other midwives irrespective of groups as designated for enrollment? If the second option is here, then there exists no difference between the groups if the follow-up was done by a third party. Result: The success rate of this procedure in this method is 94.3 % and 95.3% in physician and midwife's group [Table-3] where as 14% women experienced heavier than normal bleeding in each group[table-4]. These data evokes a query whether these 14% women in follow-up needed any extra care? Can they be designated as "failure"? If not, why? I am not going into the detailed results as well as discussion part. Those also needs to be more precise and oriented.
--	---

REVIEWER	andrzej kulczycki University of Alabama at Birmingham (UAB) USA
REVIEW RETURNED	20-Feb-2017

GENERAL COMMENTS	The paper concerns an important topic, and adds evidence that midwives can treat incomplete abortion with misoprostol. It is encouraging to see more papers on this. In the checklist, the items noted 'no' should have the appropriate modifications made in the text. These and some additional edits should be recommended and should be manageable for the authors: 1). Title: The work is done in 1 hospital (see also point 5) in 1 city in Kenya, so this must be inserted, because it is currently misleading to suggest this work was done throughout the country or that it is somehow representative of Kenya.
---

2). Abstract and Use of English there and throughout manuscript: some editing is needed throughout to improve on the writing and make it read better, e.g. the very first entry in the Abstract, 'Objective' has a second sentence that starts off with: 'Secondly to assess...'; and 'Main Outcome measures' has a second line that starts: 'Secondary, women's....'

3). Introduction: First 3 sentences all need correcting. These are all estimates.
The key thoughts are here, but they are rather disjointed and the Introduction should be strung together more coherently.

4). Top of p.5: Aims:
the first aim has seen a few studies in SSA and these should be cited. It is good that there should be more studies on the topic, but earlier work should be credited.
Second aim is listed as acceptability, but the paper does not get at this. You only have very vague questions with self-reported answers that allude to acceptability. Also, these are not analyzed (in part because they cannot be, they lack depth), they can only be described. The second aim should be revised. It can be broadened to include other secondary outcomes, but these should be presented more modestly—you only have very crude measures.

5). Methods:
p. 5, parag. 1, lines 15-18, you are describing the situation in part of the third largest city in the country, so what does 'compared to Kenya' mean? Also, we do not know the situation exactly elsewhere.
parag. 2: This parag. is too short (1 sentence only) and states that this is a multicenter RCT, but the previous parag, says it is conducted at one teaching and referral hospital. The next parag. mentions 2 hospitals, be it the pilot or main study, or both. All this needs clarification.

6). Procedures: parag. 1: Presumably, at least some of the health care providers would have been trained before to the standards. Was some allowance made for such differences in training and experience, be it in training, conduct of the study, and analysis?
parag. 2: Five contraceptive methods are listed. How does this compare to the norms at other Kenyan hospitals and to the country's national family planning program effort?
Lines 43-45: could providers have favored one method over other for reasons not controlled for, e.g. financial or reimbursement barriers, medical biases, etc. This should be acknowledged as a study limitation.
Line 44: Was follow-up of contraceptive use only done for 7-10 days after the PAC procedure? Was contraceptive use only recorded at the moment of discharge? This is another limitation.

7). Outcomes and statistical analysis:
The description on p. 8 seems in order, but:
a). the predetermined margin of equivalence (-4 to 4%) – which seems reasonable to me—should be justified
b). 'facility' is mentioned as a covariate—we do not know what this means because the study may have only been conducted at the teaching hospital.
c. line 53-4 states that 'Safety parameters of the procedure were viewed descriptively without any formal statistical testing'—ok, but please justify or state as a limitation.

8). Results, p. 9:
Losses to follow-up are adequately explained and Fig. 1, tables 1-3 are clear.
What does the statement: physicians were more experienced than the midwives in PAC" mean specifically? What sort of experience did the midwives have? e.g. were they just assisting physicians, as is

	the case in many Kenyan and other African hospitals. 9). Regarding the secondary outcomes: The authors say they have assessed acceptability, but this is not the case. The only measure given (p.10, lines 30-32) are whether the women perceived the treatment to be as expected or easier than expected, and if they would recommend it to a friend, both of which are very crude and tell us very little about acceptability. One cannot conclude that the treatment is 'highly acceptable' (line 52-54 and in the discussion and abstract), nor that contraceptive uptake is high when we only know the outcome immediately post-treatment. On p.10, several of the secondary outcomes (e.g. bleeding, pain, symptoms) are discussed under Main Outcomes, whereas in the Methods section, they are listed as secondary outcomes and they should be placed in this results sub-section. 10). Discussion The stated limitations are correct, but several more need to be added, as mentioned above, e.g. the need for long-term follow-up of contraceptive method use—certainly more than what the index woman is offered at the exit visit. p. 13, lines 22-23: there are also other studies done on other continents that should be noted with some comparison made, and South Africa's experience should also be noted.
--	--

REVIEWER	Jennifer Blum Gynuity Health Projects, USA
REVIEW RETURNED	21-Feb-2017

GENERAL COMMENTS	abstract: in population missing end date for data collection "2016" main outcome measure: its unclear to me why the medication will work differently if administered by a nurse versus a physician. What is the medical rationale for this possible difference? In the methods, the authors need to clearly state which tasks were done by nurses versus MDs and show that there was no overlap - if yes, then I am not sure what the outcome is again, is it nurse led, MD supported versus MD led, nurse supported? In either case, I don't understand the reasoning to think that the efficacy of the pills, swallowed by a woman, will be different if the pills are handed to her by one type of provider in a hospital versus another. Strengths: The first strength has already been demonstrated in a number of studies. Here are some publications to consider and add to the reference list: Glob Health Sci Pract. 2016 Sep 29;4(3):481-94. doi: 10.9745/GHSP-D-16-00052. Print 2016 Sep 28. Postabortion Care: 20 Years of Strong Evidence on Emergency Treatment, Family Planning, and Other Programming Components. Huber D1, Curtis C2, Irani L3, Pappa S4, Arrington L5. At the very top of this article in a text box the authors state: " Misoprostol, which can be provided by trained nurses and midwives, shows substantial promise for extending PAC services to secondary hospitals and primary health posts."
---

The authors should do an updated pub med review since there is a great deal published on this topic. Other pubs to review include:

Int J Gynaecol Obstet. 2014 Sep;126(3):223-6. doi: .1016/j.ijgo.2014.03.028. Epub 2014 May 15. Decentralizing postabortion care in Senegal with misoprostol for incomplete abortion. Gaye A1, Diop A2, Shochet T3, Winikoff B3.

Sublingual misoprostol versus standard surgical care for treatment of incomplete abortion in five sub-Saharan African countries. Shochet T, Diop A, Gaye A, Nayama M, Sall AB, Bukola F, Blandine T, Abiola OM, Dao B, Olayinka O, Winikoff B. BMC Pregnancy Childbirth. 2012 Nov 14;12:127. doi: 10.1186/1471-2393-12-127. - This article states : "Care throughout the study was provided by clinicians at all levels. In all sites except Mauritania and Nigeria, care was provided by midwives or nurses. In both these sites, care was mostly given by physicians although nurses and midwives were involved in counseling and follow-up"

Sublingual [corrected] misoprostol as first-line care for incomplete abortion in Burkina Faso. Blandine T, Ouattara AZ, Coral A, Hassane C, Clotaire H, Dao B, Lankoande J, Diop A, Blum J. Int J Gynaecol Obstet. 2012 Nov;119(2):166-9. doi: 10.1016/j.ijgo.2012.05.036. Erratum in: Int J Gynaecol Obstet. 2013

Introduction: I think this could be tighter and include an updated literature review. Use of terms like "high quality PAC" should be explained. They can probably also update the literature on midlevel provision of MA, or just eliminate the Nepal reference altogether from this paper (line 37.) The statement on lines 56 - 57 "Data are limited by the scarcity of RCTs and biases of the cohort studies" should be further clarified - what are the authors referring to, exactly, here.

lines 34 - 40 can be shorten for precision, probably you can eliminate info on "emancipating space for study forms"
Page 6, line 7 - please provide additional details on the potential unexpected symptoms/burden.

inclusion and exclusion - it seems like eligibility/incomplete abortion status was determined before allocation. If yes, please state which type of provider confirmed eligibility - confirming eligibility is a key task in PAC - so its helpful to know who did this.

Can the authors explain why women were asked to stay at the clinic for 4 - 12 hrs after swallowing miso? Most studies no longer do this - and it isn't common in clinical practice either.

results: I am not sure that the discussion about socio-economic status possibly or not affecting the results is a useful discussion since again, its just swallowing a few pills, why would there possibly be an effect? I suggest this section of the discuss be shortened and more concise.

Discussion: I also think the discussion could be shorter and more precise and again, would benefit from a review of relevant references. I think for the sake of recognizing the PAC consortium/community, its helpful to make clear that post abortion contraception counseling is an element of PAC, not an addition. For

	clarity the three paragraphs in the discussion starting line 18 - line 4 on the next page can probably be edited into one concise paragraph. Page 13 lines 12 - 20 can also be edited and merged into one concise paragraph on the situation in Kenya, the guideline and their potential impact on populations of women and girls throughout the country. This would leave more space for the discussion to reflect back on the results of the paper - and maybe discuss more how these services can be provided outside of tertiary level hospitals throughout Kenya (and already are) Page 13, lines 27 - 32. I think many people will disagree that there is scarce evidence that trained midwives can provide PAC. USAID recommends it and its recommended in the WHO guidelines on health worker roles in abortion care (2015) - see table page 6 and elsewhere within this document. These guidelines also should be included in the reference list - Tables: I think the total column can be eliminated in all tables, it offers no additional information. On table 2, the authors could consider merging some of the data - eg the age ranges, the # of prev pregnancies/parity could be 3 - 4 rows, etc
--	--

VERSION 1 – AUTHOR RESPONSE

Reviewer: 1

Sanjoy Kumar Bhattacharyya R G Kar Medical College, Kolkata, India

3. Please state any competing interests or state 'None declared': None declared.

ANSWER: This information is stated on line 503, page 14.

4. Title: The title of the manuscript needs to be a bit informative. Misoprostol can be applied by different routes while managing incomplete abortion. Among them, the authors here had chosen the oral route and this need to be reflected in the study-title.

ANSWER: We chose to use misoprostol, only, to keep it as short as possible. We consider other information (design and the context) to be important as well in the title. The administration (orally), is clearly written in the abstract under the sub-heading Intervention. We are awaiting the decision from the BMJ-Editor, if to add this information in the title.

5. Introduction: The introduction part is very descriptive, elaborate as well as over-informative, which are often a bit unrelated with the purpose of the study. Yes, incomplete abortion is a very commonly encountered problem and possibility of a medical management of this complication by ground level healthcare provider is of immense help. With this background, a short, brief and informative introduction is needed.

ANSWER: Thanks, we have shorten the introduction section, to make it more stringent (13 lines shorter). Page 3 and 4.

6. Methods: a. What are the diagnostic criteria of "incomplete abortion"? The authors have included "women with signs of incomplete abortion (i.e., bleeding, contractions and lower abdominal pain during pregnancy, open cervical os, and partial expulsion)". But these signs are never alone can diagnose even a pregnant state if you do not do a simple urine for pregnancy testing initially. Moreover, this is often difficult to differentiate between complete abortions with incomplete abortion

even after clinical examination. But authors here do not take the help of ultrasonography before enrolling. These points need to clarify here in details.

b. What were the basic differences in enrollment, intervention and follow-up in between two groups, while managing the incomplete abortions? Are the diagnosis and subsequent enrollment of women was done by the physician or midwife in their respective group [So that there could be a more specific enrollment which could avoid the possibility of inclusion of lots of complete abortions]?

c. Enrollment was done by two midwives initially irrespective of subsequent allocation. Thus, in this respect, these two groups are same. Are the basic intervention is technically difficult or need expertise or experience so that it might create a difference in between the groups?

d. Only a single dose of oral Misoprostol was applied in both groups and this needs neither physician's knowledge nor midwife's experience. Were the follow up done by the groups differently? Not mentioned in details. This is a very important area of this study where you need to ensure whether the woman has achieved the complete abortion. How it was ensured? The authors have mentioned few signs of untoward events and absence of them is considered as the successful outcome. But this needs to be more detailed. Moreover, who has done those follow-up? Respective groups or any other midwives irrespective of groups as designated for enrollment? If the second option is here, then there exists no difference between the groups if the follow-up was done by a third party.

ANSWER: a. Of course this is important to state, and that was such an obvious criteria that we did note state it in detail. We have added that they did a pregnancy test at the facility. It is also true that it could be difficult to sort out differential diagnoses to incomplete abortion, such as complete abortion. However, that's the true situation in many places in Kenya. Many facilities don't have an ultrasound as a standard equipment or the routine to use it. If they need to use an ultrasound they need to transfer the women to another ward (if it is available), and the woman has to pay for this additional exam. We have done some clarifications regarding inclusion and exclusion criteria. a. Line 229-235, p 6.

b. None. Four midwives at each facility were trained, two to be the head research assistants, and two to stand-in (responsible for the first screening, enrolment, the allocation and follow up at 7-10 days). Already described on page 7, however we have revised and included the two stand-in. Line 262-263, p 7, Line 271, p 7.

c. Please see response 6b. To provide the misoprostol was not considered to be a technical procedure. But, both groups went through the same training program with misoprostol, because before the current training, the superior regime to treat a first trimester incomplete abortion was MVA.

d. Please, see response no 6a-c. Misoprostol was applied in both groups and it may be true, that this needs neither physician's knowledge nor midwife's experience. However, in these settings (like in many other settings in worldwide) regulations restrict misoprostol prescription and supervision to physicians. In addition, these kind of settings lacks modern equipment's and skills how to use them, and therefore is ultrasound rarely used in this objective. Each and every country needs their own studies that applies to the unique context before new medical routines are implemented. Kenya does not differ in that context - this study has been very well implemented at the two facilities, the midwives continues (up to date) to treat first trimester incomplete (as the superior regime).

7. Result: The success rate of this procedure in this method is 94.3 % and 95.3% in physician and midwife's group [Table-3] where as 14% women experienced heavier than normal bleeding in each group [table-4]. These data evokes a query whether these 14% women in follow-up needed any extra care? Can they be designated as "failure"? If not, why?

ANSWER: Heavier than normal bleeding could not be considered as a failure for complete abortion. This is self-reported statements and was considered a normal variation. Among the women who reported heavier bleeding than normal bleeding (14%), only 15% were diagnosed with incomplete abortion (needed repeated dosage of misoprostol or MVA). 85% were complete abortion.

Reviewer: 2

Andrzej Kulczycki

University of Alabama at Birmingham (UAB), USA

8. Please state any competing interests or state 'None declared': None declared

ANSWER: Please, see response no. 3

9. The paper concerns an important topic, and adds evidence that midwives can treat incomplete abortion with misoprostol. It is encouraging to see more papers on this. In the checklist, the items noted 'no' should have the appropriate modifications made in the text. These and some additional edits should be recommended and should be manageable for the authors:

ANSWER: Please, see response no. 1

10. Title: The work is done in 1 hospital (see also point 5) in 1 city in Kenya, so this must be inserted, because it is currently misleading to suggest this work was done throughout the country or that it is somehow representative of Kenya.

ANSWER: The study was conducted at two large public hospitals, line 191-192, p 5. This area is considered to be "a" low-resource setting (mentioned in the title), similar to other areas in Kenya, also with regard to public hospital/facilities. The third largest city does not necessarily mean that it is not a low-resource setting. We can add Kisumu, if the BMJ-Editor agree to this suggestion.

11. Abstract and Use of English there and throughout manuscript: some editing is needed throughout to improve on the writing and make it read better, e.g. the very first entry in the Abstract, 'Objective' has a second sentence that starts off with: 'Secondly to assess...'; and 'Main Outcome measures' has a second line that starts: 'Secondary, women's...'

ANSWER: Thanks. We have made some adjustments in the abstract and also adjusted the English throughout the MS. Before we submitted the MS, we used Scribendi for proof editing of the language.

12. Introduction: First 3 sentences all need correcting. These are all estimates. The key thoughts are here, but they are rather disjointed and the Introduction should be strung together more coherently.

ANSWER: Please, see response no.5.

13. Top of p.5: Aims: a. The first aim has seen a few studies in SSA and these should be cited. It is good that there should be more studies on the topic, but earlier work should be credited. b. Second aim is listed as acceptability, but the paper does not get at this. You only have very vague questions with self-reported answers that allude to acceptability. Also, these are not analyzed (in part because they cannot be, they lack depth), they can only be described. The second aim should be revised. It can be broadened to include other secondary outcomes, but these should be presented more modestly—you only have very crude measures .

ANSWER: a. That's true. However we have chosen (according to reference limitations) references addressing the same topic (incomplete abortion) and similar contexts, as well as the intervention design "midwife/Physicians". When you add these limitations, there are not many studies within this objective. However, we have revised the reference list, accordingly to your comments.

b. We agree to your comments, the questions are vague. We have added this as a limitation and we have revised the objective and abstract, to put the highlight mainly on the primary objective/outcome. However, to know the acceptability among the subjects is also a very difficult objective to measure. We considered that self-reported statements was one way to find out how they perceived the treatment. Over 90% reported that they; felt safe, perceived the treatment as expected/easier as expected, and will recommend it to a friend or a relative. Since all the measures were in favors to the

treatment (without any doubts), we still consider these aspects to be a warranty for the acceptance of the treatment. Page 3 (limitations) and revised Abstract, page 2.

14. Methods: a. p. 5, parag. 1, lines 15-18, you are describing the situation in part of the third largest city in the country, so what does 'compared to Kenya' mean? Also, we do not know the situation exactly elsewhere.
- b. parag. 2: This parag. is too short (1 sentence only) and states that this is a multicenter RCT, but the previous parag, says it is conducted at one teaching and referral hospital. The next parag. mentions 2 hospitals, be it the pilot or main study, or both. All this needs clarification.
- c. Procedures: parag. 1: Presumably, at least some of the health care providers would have been trained before to the standards. Was some allowance made for such differences in training and experience, be it in training, conduct of the study, and analysis?
- d. parag. 2: Five contraceptive methods are listed. How does this compare to the norms at other Kenyan hospitals and to the country's national family planning program effort?
- e. Lines 43-45: could providers have favored one method over other for reasons not controlled for, e.g. financial or reimbursement barriers, medical biases, etc. This should be acknowledged as a study limitation.
- f. line 44: Was follow-up of contraceptive use only done for 7-10 days after the PAC procedure? Was contraceptive use only recorded at the moment of discharge? This is another limitation.
- g. outcomes and statistical analysis: The description on p. 8 seems in order, but: the predetermined margin of equivalence (-4 to 4%) – which seems reasonable to me—should be justified
- h. 'facility' is mentioned as a covariate—we do not know what this means because the study may have only been conducted at the teaching hospital.
- i. line 53-4 states that 'Safety parameters of the procedure were viewed descriptively without any formal statistical testing'—ok, but please justify or state as a limitation.

ANSWER: a. We have revised the text and added information about the country. Line 198-201, p 5, line 262-263, p 7.

b. Please see response no 10.

c. Please see response no 6a-c.

d. We have added information about how the contraception relates to Kenya. Line, 157-158, p 4, line 274-277, p 8.

e. Yes, it could be indeed. However, women seeking care in these public facilities are considered to be of low SES, and with hence pronounced financial reimbursements barriers. Our analyses showed similar distribution of provided contraceptive methods, between midwives and physicians (Table 5).

f. Yes, and this is already mentioned as a limitation (page 3, and in the discussion). The contraceptive uptake was recorded at follow-up (7-10d). Line 278, p 8

g. We have already mention a reference, line 325, page 9. However we have added one more. One additional ref added (26)

h. Please, see response no 10.

i. Please see response no 18.

15. Results, p. 9: a. Losses to follow-up are adequately explained and Fig. 1, tables 1-3 are clear. What does the statement: physicians were more experienced than the midwives in PAC" mean specifically? What sort of experience did the midwives have? e.g. were they just assisting physicians, as is the case in many Kenyan and other African hospitals.

b. Regarding the secondary outcomes: The authors say they have assessed acceptability, but this is not the case. The only measure given (p.10, lines 30-32) are whether the women perceived the treatment to be as expected or easier than expected, and if they would recommend it to a friend, both of which are very crude and tell us very little about acceptability. One cannot conclude that the treatment is 'highly acceptable' (line 52-54 and in the discussion and abstract), nor that contraceptive uptake is high when we only know the outcome immediately post-treatment.

c. On p.10, several of the secondary outcomes (e.g. bleeding, pain, symptoms) are discussed under Main Outcomes, whereas in the Methods section, they are listed as secondary outcomes and they should be placed in this results sub-section.

ANSWER: a. Please, see responses no. 6a-c. The provider's previous experiences in PAC is described in Table 1. They were asked to state their working experience as shown in the Table. No further analyses were made. The superior regime to treat incomplete abortion (first trimester was MVA in the both facilities).

b. Please see responses no 13b and 14f. Contraceptive up-take was recorded at follow-up (7-10d). Please see response no 18. We also did a follow-up at 3-month, and we are still analyzing this data and it will be presented elsewhere (PhD-stud).

c. Thanks, we have revised the structure for the outcomes. We still believe that some outcomes are better placed under main outcomes. Therefore, we have divided table 4 into two tables, (Table 4 and 5). c. Page 3, limitations and Table 5 (divided table 4 into two tables). Line 294-322, page 9

16. Discussion- The stated limitations are correct, but several more need to be added, as mentioned above, e.g. the need for long-term follow-up of contraceptive method use—certainly more than what the index woman is offered at the exit visit. p. 13, lines 22-23: there are also other studies done on other continents that should be noted with some comparison made, and South Africa's experience should also be noted.

ANSWER: We have added that the lack of long-term follow-up is a limitation. It is already mentioned that future research needs to address long-term follow up for contraceptive uptake. Please see response 17 (regarding other studies/references). We have also revised the discussion section. Line 420-421, p 12. Line 467-476, p 13-14 Line 415-416, page 12. Line 467-476, page 13-14. New reference (25) and changed references 17&18

Reviewer: 3

Jennifer Blum

Gynuity Health Projects, USA

17. Please state any competing interests or state 'None declared': None declared.

ANSWER: Please see response no 3.

18. abstract: a. in population missing end date for data collection "2016"

b. main outcome measure: its unclear to me why the medication will work differently if administered by a nurse versus a physician. What is the medical rationale for this possible difference?

c. In the methods, the authors need to clearly state which tasks were done by nurses versus MDs and show that there was no overlap - if yes, then I am not sure what the outcome is again, is it nurse led, MD supported versus MD led, nurse supported? In either case, I don't understand the reasoning to think that the efficacy of the pills, swallowed by a woman, will be different if the pills are handed to her by one type of provider in a hospital versus another.

ANSWER: a. We have added the information in the method section, and awaiting decision from BMJ-editor, if to add it in the abstract.

b. Please, see responses no 6b-c.

c. Please, see responses no 6b-c.

19. Strengths: The first strength has already been demonstrated in a number of studies. Here are some publications to consider and add to the reference list: Glob Health Sci Pract. 2016 Sep 29;4(3):481-94. doi: 10.9745/GHSP-D-16-00052. Print 2016 Sep 28. Postabortion Care: 20 Years of Strong Evidence on Emergency Treatment, Family Planning, and Other Programming Components.

Huber D1, Curtis C2, Irani L3, Pappa S4, Arrington L5. At the very top of this article in a text box the authors state: "Misoprostol, which can be provided by trained nurses and midwives, shows substantial promise for extending PAC services to secondary hospitals and primary health posts." The authors should do an updated pub med review since there is a great deal published on this topic. Other pubs to review include: Int J Gynaecol Obstet. 2014 Sep;126(3):223-6. doi: .1016/j.ijgo.2014.03.028. Epub 2014 May 15. Decentralizing postabortion care in Senegal with misoprostol for incomplete abortion. Gaye A1, Diop A2, Shochet T3, Winikoff B3.

Sublingual misoprostol versus standard surgical care for treatment of incomplete abortion in five sub-Saharan African countries. Shochet T, Diop A, Gaye A, Nayama M, Sall AB, Bukola F, Blandine T, Abiola OM, Dao B, Olayinka O, Winikoff B. BMC Pregnancy Childbirth. 2012 Nov 14;12:127. doi: 10.1186/1471-2393-12-127. - This article states : "Care throughout the study was provided by clinicians at all levels. In all sites except Mauritania and Nigeria, care was provided by midwives or nurses. In both these sites, care was mostly given by physicians although nurses and midwives were involved in counseling and follow-up"

Sublingual [corrected] misoprostol as first-line care for incomplete abortion in Burkina Faso. Blandine T, Ouattara AZ, Coral A, Hassane C, Clotaire H, Dao B, Lankoande J, Diop A, Blum J. Int J Gynaecol Obstet. 2012 Nov;119(2):166-9. doi: 10.1016/j.ijgo.2012.05.036. Erratum in: Int J Gynaecol Obstet. 2013

ANSWER: Thanks for your update. Please see our response no 13a. We have revised the reference list and the strengths and limitations, p 3.

20. Introduction: I think this could be tighter and include an updated literature review. Use of terms like "high quality PAC" should be explained. They can probably also update the literature on midlevel provision of MA, or just eliminate the Nepal reference altogether from this paper (line 37.) The statement on lines 56 - 57 "Data are limited by the scarcity of RCTs and biases of the cohort studies" should be further clarified - what are the authors referring to, exactly, here. lines 34 - 40 can be shortened for precision, probably you can eliminate info on "emancipating space for study forms"

ANSWER: Please see previous response no 5. Introduction is revised, p 3-4.

21. Method: a. Page 6, line 7 - please provide additional details on the potential unexpected symptoms/burden.

b. inclusion and exclusion- it seems like eligibility/incomplete abortion status was determined before allocation. If yes, please state which type of provider confirmed eligibility - confirming eligibility is a key task in PAC - so its helpful to know who did this.

c. Can the authors explain why women were asked to stay at the clinic for 4 - 12 hrs after swallowing miso? Most studies no longer do this - and it isn't common in clinical practice either.

ANSWER: a. Please see response no 6a-c.

b. Please see responses 6a-c.

c. Please see response 10. This was a new regime, approved at the Ministry of health specifically for this study and the providers required this procedure for safety (stay 4-12 h). Today, after the implementation, the providers does no longer keep the women for this safety-check :).

22. Results: I am not sure that the discussion about socio-economic status possibly or not affecting the results is a useful discussion since again, its just swallowing a few pills, why would there possibly be an effect? I suggest this section of the discuss be shortened and more concise.

ANSWER: Please see response no 10. There is a clear association between women with low SES and severe complications after procuring an unsafe abortion. Since this is a low-resource setting, lacking modern equipment's, routines and skills how to use them, we consider SES as a relevant

topic relevant.

23. Discussion: I also think the discussion could be shorter and more precise and again, would benefit from a review of relevant references. I think for the sake of recognizing the PAC consortium/community, its helpful to make clear that post abortion contraception counseling is an element of PAC, not an addition. For clarity the three paragraphs in the discussion starting line 18 - line 4 on the next page can probably be edited into one concise paragraph. Page 13 lines 12 - 20 can also be edited and merged into one concise paragraph on the situation in Kenya, the guideline and their potential impact on populations of women and girls throughout the country. This would leave more space for the discussion to reflect back on the results of the paper - and maybe discuss more how these services can be provided outside of tertiary level hospitals throughout Kenya (and already are). Page 13, lines 27 - 32. I think many people will disagree that there is scarce evidence that trained midwives can provide PAC. USAID recommends it and its recommended in the WHO guidelines on health worker roles in abortion care (2015) - see table page 6 and elsewhere within this document. These guidelines also should be included in the reference list -

ANSWER: We have deleted text and revised some parts, as you suggested – thanks, very helpful! Additional information, PAC-consortium is the base for this trial (used in the training program, and we have used the reference (14), and additional references are 30 & 31 (used for the PAC-training). Contraceptive counselling is part of PAC – we clearly describe the elements in PAC (Introduction). However, post-abortion contraceptive service (especially in this region), is still a crucial part of PAC. High unmet need for contraception is still a challenging work (not only because of knowledge-gap, but also because of stigmatizing attitudes surrounding contraceptive-use and its misconceptions, which is present among health care providers, stakeholders, and among publics in general). Therefore, we believe this is an element in PAC which deserves to be highlighted (not separated). Page 3-5

Page 11-14

24. Tables: I think the total column can be eliminated in all tables, it offers no additional information. On table 2, the authors could consider merging some of the data - eg the age ranges, the # of prev pregnancies/parity could be 3 - 4 rows, etc.

ANSWER: We prefer to keep the data in the current format, awaiting BMJ-editors decision.

VERSION 2 – REVIEW

REVIEWER	Sanjoy Kumar Bhattacharyya R G Kar Medical College Kolkata India
REVIEW RETURNED	08-Apr-2017

GENERAL COMMENTS	This is a manuscript rewritten to show the effectiveness of oral misoprostol to manage incomplete abortions prescribed by the mid-wives. Authors described in details the maternal morbidity related to unsafe abortion in the long introduction part. This is a study related to incomplete abortions. Then, how unsafe abortion is related to it? Do you consider incomplete abortion and unsafe abortion are same? No. These are two entirely different issues. Did you enroll the women with unsafe abortion here? Not at all and not all unsafe abortions are incomplete and vice versa. The authors randomized the subjects into two groups [ ] midwife
--

	group and physician group and showed the result is comparable in both groups. However, it is not the physician or the midwives, who enrolled or follow up the women. As the study says—"the research assistants were responsible for the screening, the follow-up visit including assessment of abortion status at follow-up and contraceptive uptake, and performing MVA when necessary". So what was the function of the two groups? Only to handover the tablet to the women? So, how a drug will act differently inside the body if given by a doctor or nurse or by anybody? One of my colleagues asked this question beforehand and I am now also raising the same issue. Then, why will the result be different? There is no need to conduct a study for this. The previous manuscript was related to the issue of contraception but the new heading does not have it as an additional study finding. However, there is a detailed description regarding the contraception inside the manuscript, which I think the authors forget to edit along with the changed title. So, I think there exist few gross errors regarding the study protocol as well as concept here. I have only mentioned some of them and not all as those are beyond correction. Thus, I am not able to accept this manuscript suitable for publication
--	--

REVIEWER	Dr. Avijit Hazra Professor Department of Pharmacology Institute of Postgraduate Medical Education & Research (IPGME&R) Kolkata, India.
REVIEW RETURNED	09-Jul-2017

GENERAL COMMENTS	The following issues may be noted / clarified to improve the report.  1. Please justify why analysis of primary outcome was done for the per protocol data set rather than the intention to treat dataset. 2. Some details of the generalized estimating equation model used to estimate the risk difference in primary outcome between the two groups is desirable. Also elaborate how the adjusted risk difference was calculated. 3. The discussion may be reorganized. Shift the second and third paragraphs (that mention the limitations and constraints) towards the end before conclusion. Introduce a short paragraph to highlight that misoprostol adverse event profile in PAC in this study matches the known adverse profile of the drug and there were no striking differences between the two categories of providers in this regard. Probably a more immediate future research need is to see if the results of this study are replicable in a primary health care setting, which (as the authors themselves highlight) is somewhat different from a district hospital setting.  4. Table 3 information can be succinctly incorporated in the main text and this table omitted. The adverse event profile listed in Appendix 1 can instead be brought in as a table since safety aspect
--

	is always important in any pharmacological intervention. 5. Page 2, Line 33 Correct the timeline in the abstract. 6. Page 3, Lines 12-13 The very nature of the outcome measure requires that the data would be captured from self-reported statements during the follow-up visits. Therefore this is not a limitation. Instead you may include the point that both midwives and physicians were working in the same health facility and therefore there could have been interaction between the two sets of providers in post-abortion care. 7. Page 4, Line 37 Change 'effective' to 'effectively' 8. Page 4, Line 51 Omit the superlative term 'overarching' 9. Page 5, Line 19 Rather than stating a single figure of 'around 26', provide the applicable monthly range. 10. Page 5, Line 44 Expand on the acronyms JOOTRH and KCH at first occurrence. 11. Page 5, Line 56 Change 'obtained' to 'experienced'. 12. Page 6, Line 45-52 Do you mean 'protocol' or 'case report form' in this paragraph? 13. Page 7, Line 18-19 Provide generic composition of Depo-Provera and close bracket. 14. Page 8, Line 55-58 The equivalence margin has already been stated earlier. No need to repeat again. Please delete. 15. Page 9, Line 25-38 Flow of study participants has been presented appropriately in Figure 1. No need to repeat here. 16. Page 10, Lines 23-24 It is preferable to substantiate that there were no differences in the secondary outcomes between the Groups by adding a column of p value for intergroup comparison in Table 4 and Table 5. 17. Page 11, Line 3 Instead of writing 'The study design strengthen the reliability and the validity by the sample size ...'. You may simply write 'The strengths of the study lie in its large sample size ...'. 18. Page 13, Lines 12-16 The unsafe abortion consequences are not particularly relevant to discussion of the results of this study. May omit this. Comments concluded
--	---

VERSION 2 – AUTHOR RESPONSE

Reviewer: 1

Sanjoy Kumar Bhattacharyya

R G Kar Medical College, Kolkata, India

Please state any competing interests or state 'None declared': None declared

Please leave your comments for the authors below

This is a manuscript rewritten to show the effectiveness of oral misoprostol to manage incomplete abortions prescribed by the mid-wives.

Authors described in details the maternal morbidity related to unsafe abortion in the long introduction part. This is a study related to incomplete abortions. Then, how unsafe abortion is related to it? Do you consider incomplete abortion and unsafe abortion are same? No.

These are two entirely different issues. Did you enroll the women with unsafe abortion here? Not at all and not all unsafe abortions are incomplete and vice versa.

The authors randomized the subjects into two groups◊ midwife group and physician group and showed the result is comparable in both groups. However, it is not the physician or the midwives, who enrolled or follow up the women. As the study says—“the research assistants were responsible for the screening, the follow-up visit including assessment of abortion status at follow-up and contraceptive uptake, and performing MVA when necessary”. So what was the function of the two groups? Only to handover the tablet to the women? So, how a drug will act differently inside the body if given by a doctor or nurse or by anybody? One of my colleagues asked this question beforehand and I am now also raising the same issue. Then, why will the result be different? There is no need to conduct a study for this.

The previous manuscript was related to the issue of contraception but the new heading does not have it as an additional study finding. However, there is a detailed description regarding the contraception inside the manuscript, which I think the authors forget to edit along with the changed title.

So, I think there exist few gross errors regarding the study protocol as well as concept here. I have only mentioned some of them and not all as those are beyond correction. Thus, I am not able to accept this manuscript suitable for publication

ANSWER: Thanks for your comments. We have now clarified how unsafe abortion and PAC is related (please see our revised Introduction). Unsafe abortion is the reasons why the term post-abortion care exists. Many women who have made an unsafe abortion suffer from complications and are thus in need of post-abortion care (PAC). PAC is a global public health issue and a finance burden for the health care systems. In 2014 the cost for providing PAC was estimated to \$232 million, but it was further concluded that if all abortions had occurred under safe conditions, this cost would be reduced to \$20 million. <https://www.guttmacher.org/report/adding-it-costs-and-benefits-investing-sexual-and-reproductive-health-2014>

It is, however, not possible to separate complications due to miscarriage from complications due to unsafe abortion, which is well known within this research field (in both cases, all fetal or placental tissue have not been expelled and severe bleeding, or infection occur and prompt medical attention is required). Statistics regarding proportion of safe/unsafe abortion doesn't exist in these contexts.

Women do not usually share information whether they have tried to induce the abortion by themselves or with help from someone else. The reason being that abortion in this context is consider a crime, and the woman (and the provider) risking social and legal punishment if she admits conducting an induced abortion. This may help to explain why robust data/statistics are lacking in this objective. Estimates on (unsafe) abortion are calculated on the number of women seeking PAC. In addition, treatment for complications due to early miscarriage or early unsafe abortion is somewhat

the same. Therefore, in the introduction it is explained that “most” abortions in Kenya are conducted outside health care facilities and are therefore considered unsafe. Meaning, most women in this study had obtained an unsafe abortion (now further explained in our revised Introduction, p 15-18). Another obvious fact (in the current study) supporting this assumption, is the high proportion who accepted to start a contraceptive method. The most common contraceptive choice was a long acting reversible method– supporting the statement that pregnancy (including the current) is not desired. Please, also see the reference list, studies within this field and in similar contexts are published in prominent journals (without robust statistics). However, we have now tried to clarify this in the Introduction in accordance to your kind comments.

It is a misunderstanding that the research assistants were separated from the two groups. The research assistants were trained midwives. These research assistants were part of the staff who conducted the study, but were also the ones who did the followed up of patients (7-10 day after treatment provided by a midwife or a physician).

It is stated in the method section that 4 midwives in the staff were trained to be research assistants, with the aim to overview the work, and conduct the follow-up. We have made some revisions in the text to make this clearer, as we understand that this was not clear, p 8, line 10-16.

According to contraception. In the trial registration contraception was registered as “Other measures”. PAC-concept includes contraceptive counseling, and by default this finding should be in the result, but not necessary in the title.

Reviewer: 4

Dr. Avijit Hazra

Department of Pharmacology

Institute of Postgraduate Medical Education & Research (IPGME&R). Kolkata, India.

Please state any competing interests or state ‘None declared’: None declared.

Please leave your comments for the authors below

Manuscript ID: bmjopen-2017-016157.R1

Post-abortion care with misoprostol - equally effective, safe and accepted when administered by midwives compared to physicians: a randomised controlled equivalence trial in a low-resource setting in Kenya.

Comments from Referee

The authors have reported the results of a randomized controlled trial of oral misoprostol in post-abortion care used by midwives in comparison to physicians and have concluded that there is equivalence of the results in terms of the primary outcome of complete abortion rate achieved. The secondary outcomes related to consumer acceptability were also comparable. Although the study is restricted to a particular region of Kenya, the sample size is adequate and the methodology is sound. Therefore the results should be representative of other resource constrained settings where the population catered to mostly hail from weak socioeconomic background.

The following issues may be noted / clarified to improve the report.

ANSWER: Thank you for constructive and helpful comments.

1. Please justify why analysis of primary outcome was done for the per protocol data set rather than the intention to treat dataset.

ANSWER: The intention to treat (ITT) shows all the women randomized into the study with the aim to be treated. Therefore, the characteristics of the participants is presented on the ITT group (Table 2), which is important since it can provide information about the randomisation, whether it's correctly done or not. However, our outcomes are measured at follow-up and we lack data about the participants lost to follow-up. The outcomes are therefore analysed by per protocol and not by the ITT group. Sub-analyses were done on the group lost to follow-up (comparisons to the follow-up group and by provider group), and presented in the MS. Revised, p 9, line 26-28.

2. Some details of the generalized estimating equation model used to estimate the risk difference in primary outcome between the two groups is desirable. Also elaborate how the adjusted risk difference was calculated.

ANSWER: The margin was based on clinically and statistically important differences as well as ethical criteria, cost, and feasibility shown in previous studies, which we have now added. P 9, line 21-23.

3. The discussion may be reorganized. Shift the second and third paragraphs (that mention the limitations and constraints) towards the end before conclusion.

ANSWER: Thanks for the suggestion; we have now re-organized the discussion accordingly.

Introduce a short paragraph to highlight that misoprostol adverse event profile in PAC in this study matches the known adverse profile of the drug and there were no striking differences between the two categories of providers in this regard.

ANSWER: Thanks for advice, we have added this information, p 14, line 9-10.

Probably a more immediate future research need is to see if the results of this study are replicable in a primary health care setting, which (as the authors themselves highlight) is somewhat different from a district hospital setting.

ANSWER: Thanks, we have added this suggestion, p 14, line 16-18.

4. Table 3 information can be succinctly incorporated in the main text and this table omitted. The adverse event profile listed in Appendix 1 can instead be brought in as a table since safety aspect is always important in any pharmacological intervention.

ANSWER: Since this is the main-outcome and the presentation follow the consort-guidelines we decided to keep this format, without changes (however, we are willing to reconsider if this is the wish of the editor)

5. Page 2, Line 33

Correct the timeline in the abstract.

ANSWER: Corrected, June 1th, 2013 until May 31, 2016.

6. Page 3, Lines 12-13

The very nature of the outcome measure requires that the data would be captured from self-reported statements during the follow-up visits. Therefore this is not a limitation. Instead you may include the point that both midwives and physicians were working in the same health facility and therefore there could have been interaction between the two sets of providers in post-abortion care.

ANSWER: We have followed your suggestion and made changes in the text accordingly, p 3, line 11-12.

7. Page 4, Line 37

Change 'effective' to 'effectively'

ANSWER: Done, p 5, line 16.

8. Page 4, Line 51

Omit the superlative term 'overarching'

ANSWER: Done, p 5, line 22.

9. Page 5, Line 19

Rather than stating a single figure of 'around 26', provide the applicable monthly range.

ANSWER: Done, p 6, line 17.

10. Page 5, Line 44

Expand on the acronyms JOOTRH and KCH at first occurrence."

ANSWER: This was already done, p 6, line 4-5.

11. Page 5, Line 56

Change 'obtained' to 'experienced'.

ANSWER: Done, p 6, line 29.

12. Page 6, Line 45-52

Do you mean 'protocol' or 'case report form' in this paragraph?

ANSWER: We mean case report form, and that the report form also contains protocols for each woman, p 7, line 22.

13. Page 7, Line 18-19

Provide generic composition of Depo-Provera and close bracket.

ANSWER: Done, p 8, line 12.

14. Page 8, Line 55-58

The equivalence margin has already been stated earlier. No need to repeat again. Please delete.

ANSWER: Done, p 10, line 1-3.

15. Page 9, Line 25-38

Flow of study participants has been presented appropriately in Figure 1. No need to repeat here.

ANSWER: Revised, p 10, line 16-22.

16. Page 10, Lines 23-24

It is preferable to substantiate that there were no differences in the secondary outcomes between the Groups by adding a column of p value for intergroup comparison in Table 4 and Table 5.

ANSWER: These tables are descriptive data and we added p-value on contraceptive uptake p 11, line 27-30, however we don't believe P-values will add anything in the tables. N, %, M, SD are presented and the result speaks by itself. The text concludes the table-result and states the non-significant information (p 11, line 20-21), between the two groups. According to Consolidated Standards of Reporting Trials (CONSORT), endorsed by over 50% of the PubMed journals, state that significance tests should be avoided in descriptive tables <http://www.statisticalmisses.nl/index.php/frequently-asked-questions/84-why-are-significance-tests-of-baseline-differences-a-very-bad-idea>

17. Page 11, Line 3

Instead of writing 'The study design strengthens the reliability and the validity by the sample size ...'.

You may simply write 'The strengths of the study lie in its large sample size ...'.

ANSWER: Thanks, done, p 13, line 22.

18. Page 13, Lines 12-16

The unsafe abortion consequences are not particularly relevant to discussion of the results of this

study. May omit this.

ANSWER: Done, p 14, line 21-24.